# Enhancing Multilingual LLM Pretraining with Model-Based Data Selection

**Bettina Messmer**[*]
EPFL
bettina.messmer@epfl.ch

**Vinko Sabolčec**[*]
EPFL
vinko.sabolcec@epfl.ch

**Martin Jaggi**
EPFL
martin.jaggi@epfl.ch

## Abstract

Dataset curation has become a basis for strong large language model (LLM) performance. While various rule-based filtering heuristics exist for English and multilingual datasets, model-based filtering techniques have primarily focused on English. To address the disparity stemming from limited research on non-English languages, we develop a model-based filtering framework for multilingual datasets that aims to identify a diverse set of structured and knowledge-rich samples. Our approach emphasizes transparency, simplicity, and efficiency, leveraging Transformer- and FastText-based classifiers to ensure the broad accessibility of our technique and data. We conduct comprehensive ablation studies on the FineWeb-2 web crawl dataset across diverse language families, scripts, and resource availability to demonstrate the effectiveness of our method. Training a 1B-parameter Llama model for 70B and 119B tokens, our approach can match the baseline MMLU score with as little as 15% of the training tokens, while also improving across other benchmarks and mitigating the curse of multilinguality. These findings provide strong evidence for the generalizability of our approach to other languages. As a result, we extend our framework to 20 languages for which we release the refined pretraining datasets.

## 1 Introduction

Large Language Models (LLMs) have demonstrated impressive performance improvements when trained on increasingly larger datasets and model sizes [Brown et al., 2020]. While Brown et al. [2020] already observed the importance of using a cleaned version of Common Crawl for improved performance, the high cost of LLM training has further motivated research into better pretraining quality filters.

Deduplication and heuristic-based dataset cleaning have become standard practices in data curation [Rae et al., 2021, Raffel et al., 2020, De Gibert et al., 2024]. These quality filters are often complemented by additional filters, such as the removal of personally identifiable information (PII) [Penedo et al., 2024a] or model-based toxicity filtering [Soldaini et al., 2024]. Recently, model-based filtering has also emerged as a promising method for quality filtering. The release of FineWeb-Edu [Penedo et al., 2024a] demonstrated that pretraining on just 10% of the tokens (38B) from an English dataset filtered using a model-based approach can achieve performance comparable to models trained on 350B tokens of unfiltered data. Moreover, when trained on equivalent amounts of data, this model largely outperforms the baseline. Concurrently, the release of DataComp-LM (DCLM) [Li et al., 2024b] showed that competitive performance can be achieved using a simple and efficient model-based approach, namely a FastText [Joulin et al., 2017] classifier trained on a carefully selected training dataset.

---

[*]Equal contribution

39th Conference on Neural Information Processing Systems (NeurIPS 2025) Track on Datasets and Benchmarks.

However, these recent advances have primarily focused on English data. This emphasis risks further widening the disparity in LLM performance between languages, as less than half of internet content is written in English[2]. To address this concern, we aim to extend model-based filtering frameworks to multilingual datasets. While model perplexity-based filtering is commonly applied to multilingual datasets [Wenzek et al., 2019, Laurençon et al., 2022, Nguyen et al., 2023], the current state-of-the-art, FineWeb-2 [Penedo et al., 2024c], primarily relies on heuristic-based filters. In this work, we focus on model-based filtering with a quality definition that emphasizes: 1) structured data and 2) knowledge-rich data samples, to enhance multilingual pretraining datasets.

To achieve this, we leverage embedding-based classification models. Firstly, we adopt the FastText quality filtering approach from DCLM to develop a unified framework for multilingual datasets that span diverse language families, scripts, and resource availability, focusing on Chinese, German, French, Arabic, and Danish as representative languages for our experiments. Additionally, we extend this embedding-based approach by incorporating Transformer [Vaswani et al., 2023] embeddings, specifically XLM-RoBERTa [Conneau et al., 2020], for filtering. Figure 1 shows the clear performance gains of our best FastText and Transformer embedding-based approaches over the state-of-the-art baseline FineWeb-2 data.

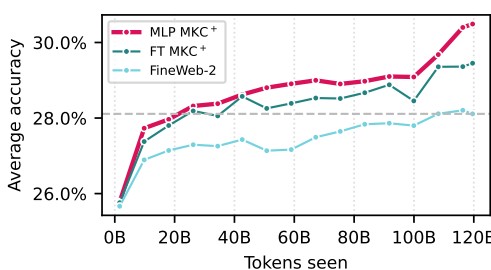

Figure 1: Average accuracy on Chinese (CMMLU), German (MMLU), and French (MMLU) benchmarks during training: FineWeb-2 baseline compared to our methods (10% data retention).

In summary, our contributions are as follows:

- We develop a transparent, simple, and unified framework for multilingual model-based filtering at web scale, enabling data curation across diverse language families, scripts and resource availability.
- We present comprehensive per-language ablation studies of embedding-based multilingual quality filtering on top of the FineWeb-2 dataset [Penedo et al., 2024c], achieving performance comparable to the baseline while using as little as 15% of the tokens. We additionally analyze the impact of dataset contamination. Lastly, our experiments show that our dataset doesn't suffer from the *curse of multilinguality* [Chang et al., 2023].
- We evaluate the impact of different data selection classifiers, in particular their training datasets, on the downstream performance of LLMs.
- We release the refined pretraining dataset[3] covering 20 languages[4], filtered using our proposed framework, along with the codebase[5], to advance multilingual language modeling.

## 2  Related Work

**Data Curation.** In order to pretrain LLMs on a large amount of diverse texts, Common Crawl[6] is often used as the base dataset. However, early works already observed that performing data curation on Common Crawl is crucial for model performance [Brown et al., 2020]. In fact, data curation is important across NLP tasks [Peter et al., 2023, Finkelstein et al., 2024]. Specifically for pretraining data, there exist various data curation approaches, such as deduplication [Lee et al., 2022], PII removal [Subramani et al., 2023], or toxicity filtering [Arnett et al., 2024]. Another important aspect is quality filtering of the documents. For this, the definition of quality is an important aspect. A common approach is to use heuristics to remove documents outside of the target distribution, such as filtering based on average word length, existence of punctuation, or document length [Rae et al., 2021, Raffel et al., 2020]. Another approach is to define model-based filters, where research has focused on perplexity measure of the text [Wenzek et al., 2019, Marion et al.,

---

[2]w3techs.com/technologies/overview/content_language

[3]huggingface.co/datasets/epfml/FineWeb2-HQ

[4]Russian, Chinese, German, Japanese, Spanish, French, Italian, Portuguese, Polish, Dutch, Indonesian, Turkish, Czech, Vietnamese, Swedish, Persian, Arabic, Greek, Danish, Hungarian (dataset details in Appendix A)

[5]github.com/epfml/fineweb2-hq

[6]commoncrawl.org

2023, Ankner et al., 2024], distributional similarity measures [Li et al., 2024b] and LLM-based quality assessment [Gunasekar et al., 2023, Wettig et al., 2024, Sachdeva et al., 2024, Penedo et al., 2024a]. In this work, we build upon previous curated datasets based on heuristic filtering, namely the state-of-the-art dataset FineWeb-2 [Penedo et al., 2024c], and focus on model-based filtering for structured and knowledge-rich documents relying on textual embeddings.

**Curated English datasets.** One of the early curated datasets was C4 [Raffel et al., 2020], followed by MassiveText [Rae et al., 2021]. RefinedWeb [Penedo et al., 2023] was an important step forward, demonstrating that filtered web data can outperform selected high-quality data sources. Although these datasets have not been made fully publicly available, their filtering techniques have been expanded upon in recent fully public datasets, such as Dolma [Soldaini et al., 2024], FineWeb, FineWeb-Edu [Penedo et al., 2024a] and DCLM [Li et al., 2024b]. While FineWeb primarily relies on filter heuristics for data quality, Dolma adopts model perplexity filtering. FineWeb-Edu takes model-based filtering a step further and relies on LLM-based quality assessment. DCLM, a concurrent work, has achieved competitive performance using a FastText [Joulin et al., 2017] classifier trained on a carefully selected training dataset. In this work we adapt and extend this approach to the multilingual context.

**Curated Multilingual Datasets.** Analogously to English datasets, significant work has been done in the multilingual space. For example, CCNet [Wenzek et al., 2019] has been influential, with its language identification and model perplexity filtering for data quality being adopted in subsequent datasets. Similar to earlier English datasets, CCNet was not published directly, but rather provided tools for data cleaning. RedPajama [Together Computer, 2023] is a prominent multilingual dataset relying on these filtering techniques, offering data in 5 European languages. Other datasets, such as OSCAR [Ortiz Suárez et al., 2019, Abadji et al., 2021, Abadji et al., 2022], mC4 [Xue et al., 2021], ROOTS [Laurençon et al., 2022], MADLAD-400 [Kudugunta et al., 2023], CulturaX [Nguyen et al., 2023], and HPLT [de Gibert et al., 2024], expanded coverage across a variety of language families and scripts. These datasets offer refined content for hundreds of languages, while FineWeb-2 [Penedo et al., 2024c] pushes the limit to thousands of languages and further improves performance. A concurrent work by Martins et al. [2025] uses translation to train a multilingual quality filter based on the English FineWeb-Edu [Penedo et al., 2024a] scores. Our work similarly focuses on filtering high-quality samples across various language families and scripts. However, we take a different approach: we train a separate classifier for each language from scratch using structured and knowledge-rich representative samples. We limit our scope to 20 languages as the number of documents drops quickly and there is trade-off between retaining a sufficient number of pretraining tokens and ensuring data quality [Muennighoff et al., 2023, Held et al., 2025].

**Multilingual Embedding Models.** Early word embedding models like Word2Vec [Mikolov et al., 2013] and GloVe [Pennington et al., 2014] lacked contextual understanding. FastText [Bojanowski et al., 2017] built upon them and improved performance by incorporating subword information. Transformer [Vaswani et al., 2023] models like BERT [Devlin et al., 2019] and GPT [Radford et al., 2018] then revolutionized the field with context-aware embeddings. Multilingual models like mBERT, XLM [Lample and Conneau, 2019], and XLM-RoBERTa [Conneau et al., 2020] further advanced cross-lingual understanding, with recent open-source LLMs pushing performance even higher [Llama Team, 2024, Mistral AI, 2025]. Using such Transformer models, documents and representative samples can be mapped into a shared embedding space to estimate their similarity. Focusing on transparency, simplicity and efficiency in our work, we use FastText and XLM-RoBERTa for our model-based filtering.

**Multilingual Evaluation.** Evaluating LLMs requires diverse benchmarks testing linguistic and cognitive abilities like reading comprehension, reasoning, and knowledge. While established benchmarks such as MMLU [Hendrycks et al., 2020] and ARC [Clark et al., 2018] exist for English evaluation, assessments in other languages often rely on translations from English sources, as seen in XNLI [Conneau et al., 2018] and the machine-translated version of MMLU [Lai et al., 2023]. However, translations can be problematic, failing to capture cultural nuances or introducing "translationese" [Romanou et al., 2024]. Recent work by Romanou et al. [2024] and Singh et al. [2024a] emphasizes the importance of culturally sensitive, natively collected benchmarks. Task difficulty and formulation also impact model performance when trained for shorter durations [Kydlíček et al., 2024]. In our work, we follow FineTasks, a recent evaluation tasks suite by Kydlíček et al. [2024] to assess our model-based filtering approaches across multiple languages.

# 3 Methods

In this work, we present our model-based filtering approaches. Our methodology is structured into two key components: 1) we select suitable training datasets, aiming to identifying a diverse set of structured and knowledge-rich samples and 2) we describe the different models, namely FastText and Transformer embedding-based filters, used to capture and leverage these characteristics.

## 3.1 Classifier Training Dataset

**Representative Sample Selection.** Our goal is to identify a diverse set of structured and knowledge-rich samples, especially within a multilingual context. We define two criteria for our training datasets: 1) the samples must be informative and well-structured and 2) the datasets must be available in multiple languages. While some multilingual benchmark datasets meet these criteria precisely, it is important to note that we do not train the LLM directly on this data. Instead, we train a proxy model to assess pretraining data quality. Nevertheless, we must remain cautious about potentially increased pretraining data contamination stemming from this approach, as discussed in Section 4.2.5.

Based on our criteria, we selected the following datasets as representative examples.

- *Aya Collection*. A prompt completion dataset comprising ∼514M samples covering a variety of tasks, generated using instruction-style templates in 101 languages [Singh et al., 2024b].
- *Aya Dataset*. Human-annotated instruction fine-tuning dataset consisting of ∼202K prompt-completion pairs in 65 languages [Singh et al., 2024b].
- *MMLU*. Dataset contains ∼14K multiple-choice knowledge questions on various topics in English [Hendrycks et al., 2020]. Multilingual version was translated into 14 languages by professional translators [OpenAI, 2024].
- *OpenAssistant-2*. The dataset contains ∼14K user-assistant conversations with multiple messages in 28 languages [Fischer et al., 2024].
- *Include-Base-44*. Multiple-choice questions focused on general and regional knowledge, extracted from academic and professional exams. Spanning 44 languages, it includes a total of ∼23K samples [Romanou et al., 2024].

**Representative Sample Collection.** *MMLU* and *Include-Base-44* are highly curated benchmark datasets, containing structured, knowledge-rich samples. The *Aya Dataset* is human-curated, while *OpenAssistant-2* is partially human-curated and partially generated by large language models (LLMs). In contrast, the *Aya Collection* consists of various AI-generated samples without quality guarantee, though it represents the largest and most multilingual of the five.

To address the quality difference, we create two *Multilingual Knowledge Collection (MKC)* configurations which allow us to evaluate the trade-off between data quality and scale:

- *MKC*: Includes *Include-Base-44*, *OpenAssistant-2*, *MMLU*, and the *Aya Dataset*
- $MKC^+$: Includes *MKC* and the *Aya Collection*

**Dataset Creation.** For our model-based filtering approaches, our goal is to identify documents from the pretraining dataset that are most similar to our representative samples, with the notion of similarity determined by the specific classifier used. We can directly measure similarity to our training data, for example, by calculating cosine similarity with training samples in the embedding space. Alternatively, following the approach of Li et al. [2024b], the task can be framed as a binary classification problem, with the representative samples as the positive class. For the negative class, we can subsample documents from our pretraining dataset, under the assumption that the majority of these documents are not well-structured or knowledge-rich. We use both approaches for our classifiers.

To create the binary classification training dataset, we selected up to 80K positive samples by using all examples from the smaller source datasets (e.g., *Include-Base-44*) and randomly subsampling from the *Aya Collection* for $MKC^+$. The positive samples were combined with the same number of randomly sampled negative examples from FineWeb-2. The same training dataset was utilized across all model-based filtering approaches, disregarding negative samples when unnecessary. Additionally, we created a training dataset for each language individually to avoid leaking language-specific biases to data of other languages.

**Sample Pre-processing.** We applied no pre-processing to the FineWeb-2 (negative) samples but performed minimal pre-processing on the representative (positive) samples. For instance, in datasets like *MMLU* or *OpenAssistant-2*, we concatenated various sample components. For the *Aya Collection*, we resolved encoding issues in non-Latin languages and removed samples containing *<unk>* tokens, which were particularly prevalent in Arabic data (37.1%).

### 3.2 FastText-based Filtering (FT)

To efficiently process datasets with over 100 million documents [Penedo et al., 2024c], similar to DCLM [Li et al., 2024b], we used a binary FastText classifier [Joulin et al., 2017]. FastText runs on CPU and can be deployed across multiple cores, for example using DataTrove [Penedo et al., 2024b].

We trained our FastText classifier on the processed training set using 2-gram features (4-gram for Chinese). These classifiers were then used to assign scores to all documents in the pretraining dataset. To filter the dataset, we applied a score threshold based on the desired retention percentage of documents. This approach balances dataset size and the predicted quality of the samples.

### 3.3 Transformer Embedding-based Filtering

To leverage rich semantic information based on contextual relationships, we utilized Transformer model embeddings. Specifically, we selected a pretrained XLM-RoBERTa base model [Conneau et al., 2020] due to its support of 100 languages, a relatively small size of 279M parameters, and its transparent training procedure. This choice enabled us to process web-scale data efficiently without being restricted to a single language and aligned with our commitment to open science.

To retain general embeddings that can be reused across methods, we opted against fine-tuning the model. For each document from our datasets, we computed the 768-dimensional embedding by mean pooling the embeddings of the output sequence. Since the model has a fixed maximum sequence length of 512 tokens, we considered only the first 512 tokens of each document, assuming they are representative of the entire document.

After computing the embeddings of our corpora, we experimented with two methods: 1) classification of embeddings using a multi-layer perceptron and 2) cosine similarity between the embeddings. As in the FastText approach, we scored each document and applied a threshold to retain the desired percentage of the highest-scoring documents.

**Multi-Layer Perceptron (MLP).** We trained a single-hidden-layer neural network with a dimension of 256, the ReLU activation function, a 20% dropout, and the sigmoid function on the output. The network was trained for 6 epochs using the AdamW optimizer [Loshchilov and Hutter, 2019] with a constant learning rate 0.0003 and binary cross-entropy loss. We computed document scores using the output layer of the MLP model, which used XML-RoBERTa document embeddings as input.

**Cosine Similarity (CS).** We computed the document scores as the maximum cosine similarity between its embeddings and a set of $K$ randomly sampled positive sample embeddings. We experimented with varying values of $K$, including 1024, 2048, 4096, 8192, and 16384. However, we did not observe a significant differences in the documents with high scores across these variations when manually inspecting the data. To strike a balance between the diversity of the positive samples and computational efficiency, we chose $K = 8192$ for our experiments.

## 4 Experiments

### 4.1 Experimental Setup

**Technical Details.** We evaluate 1B-parameter Llama models [Llama Team, 2024] to demonstrate the effectiveness of our model-based filtering approaches. The models are trained on either 70B or 119B tokens, balancing token quality and diversity. The smaller dataset (70B tokens) exposes the model to each token at most once (with a few exceptions where some tokens appear twice). The larger dataset (119B tokens) simulates longer training, resulting in increased token repetition. Training utilizes the HuggingFace Nanotron library [Hugging Face, 2024a] with the AdamW optimizer [Loshchilov and Hutter, 2019] and a WSD learning rate schedule [Hägele et al., 2024].

To minimize the need for costly hyperparameter tuning, we maintain a consistent setup across all experiments. Specifically, we adopt the DeepSeek scaling law [DeepSeek-AI et al., 2024] with a batch size of 1.6M tokens, learning rate of 0.0008, and 2000 warmup steps.

As the base dataset, we use FineWeb-2 [Penedo et al., 2024c], which has been shown to provide a strong baseline across a variety of languages. Since FineWeb-2 is globally deduplicated, we rehydrate both filtered and unfiltered data using the hyperparameters recommended by Penedo et al. [2024c].

To validate our method on English, we use three datasets: FineWeb [Penedo et al., 2024a] as the baseline, along with FineWeb-Edu [Penedo et al., 2024a] and DCLM [Li et al., 2024b], both of which represent the current state-of-the-art. Tokenization is performed using the multilingual Mistral v3 (Tekken) tokenizer [Mistral AI, 2024].

**Evaluation.** Our evaluation prioritizes a diverse range of tasks to ensure the models retain well-rounded capabilities, rather than focusing exclusively on knowledge-based tasks. Specifically, we include tasks covering reading comprehension, general knowledge, natural language understanding, common-sense reasoning, and generative tasks in the target language. To evaluate our approach, we use the HuggingFace LightEval library [Fourrier et al., 2023].

For French, Chinese, and Arabic, we utilize the FineTasks [Kydlíček et al., 2024] multilingual evaluation suite, which is designed to provide meaningful signals even for models trained in the order of 100B tokens. We select analogous tasks for German and Danish. For English, we rely on the SmolLM tasks suite [Hugging Face, 2024b]. A complete list of tasks and their evaluation metrics for each language is provided in Appendix D.

**Model Selection.** We follow the approach used in FineTasks [Kydlíček et al., 2024] for filter selection, computing a global rank score across individual metrics and languages to determine the optimal approach. For a detailed description of the average rank computation, please refer to Appendix E.

**Computational Cost.** We run our experiments on a compute cluster containing four GH200 chips per node, with each GH200 chip containing 72 CPU cores and one H100 GPU. Model training on 119B tokens costs approximately 1.1K H100 compute hours, while the embedding computation of all data for 20 languages costs approximately 4K H100 compute hours. We release the embeddings publicly so this computation does not have to be repeated[7]. In total, we use approximately 152K H100 compute hours for running our experiments and generating the document embeddings. Data selection classifier training (i.e. FastText and MLP) takes a few minutes on a CPU. Filtering using any of the approaches (i.e., FastText, MLP, or CS) is computationally inexpensive, parallelizable, and is run on CPU. Overall, filtering German, Chinese, French, Arabic, and Danish data for one filtering approach in the ablations costs approximately 60 CPU hours, which is distributed over multiple CPU cores depending on the dataset size in each language to have filtering finish in approximately 30 minutes.

## 4.2 Experimental Results & Discussion

### 4.2.1 Model Selection

In Section 3, we introduced several model-based filtering approaches. *But which of these performs the best?* We evaluate which combination of our defined classifier training datasets (*MKC* or *MKC$^+$*) and filtering methods (*FT*, *MLP* or *CS*) achieve the highest performance. Table 1 presents the overall ranking across our representative language selection (Chinese, German, French, Arabic, Danish) and training runs of 70B and 119B tokens. Analogous to the DCLM filtering recipe [Li et al., 2024b], the results are based on a dataset that retains 10% of the documents for the high-resource datasets (Chinese, German, French) and keeps 56% and 65% of the documents for the lower-resource languages

Table 1: Benchmark performance comparison: Average rank between FineWeb-2 baseline and our proposed filtering methods (*FT*, *MLP*, and *CS*) trained on *MKC$^+$* or *MKC*, retaining top 10% for Chinese, German, and French, 56% for Arabic, and 65% for Danish. The average rank is computed across FineTasks for 1B-parameter models evaluated after 70B and 119B tokens.

| Approach | Average Rank |
|---|---|
| *MLP MKC$^+$* | 4.35 |
| *MLP MKC* | 6.11 |
| *FT MKC$^+$* | 7.17 |
| *FT MKC* | 8.04 |
| *CS MKC* | 8.10 |
| Baseline | 8.72 |
| *CS MKC$^+$* | 8.79 |

---

[7]huggingface.co/datasets/epfml/FineWeb2-embedded

(Arabic and Danish, respectively). These percentages maintain approximately 70B tokens, under the assumption of uniform token distribution across documents. We also exclude approaches that use *MKC* for training on Danish, as it lacks sufficient training data. For detailed, per-language results, please refer to Appendix C.1.

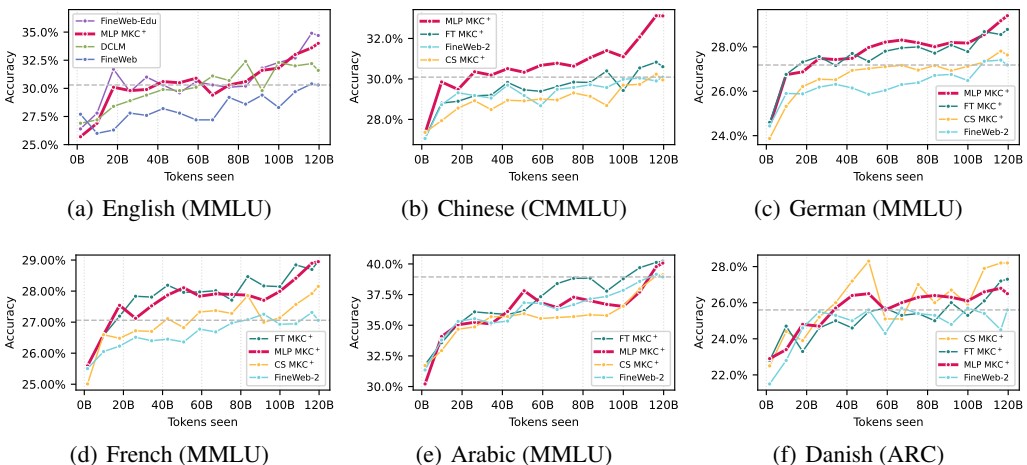

(a) English (MMLU)      (b) Chinese (CMMLU)      (c) German (MMLU)

(d) French (MMLU)      (e) Arabic (MMLU)      (f) Danish (ARC)

Figure 2: Benchmark performance comparison: Accuracy during 119B token training between baseline methods (FineWeb, DCLM, FineWeb-Edu, FineWeb-2) and our proposed filtering approaches (*FT*, *MLP*, and *CS*), trained on $MKC^+$. Our approaches use 10% data retention for English, Chinese, German, and French, 56% for Arabic, and 65% for Danish. For English, Chinese, German, and French, baseline-level performance is reached at approximately 20B tokens (16.7% of total).

Table 1 demonstrates that *MLP MKC$^+$* approach outperforms all other approaches. Interestingly, the high- and low-scored samples presented in Appendix F align with the observed rankings. Figure 2 further highlights the strong performance of *MLP MKC$^+$*, particularly for high-resource languages, where it largely outperforms the baseline. For lower-resource languages—where less data was filtered—the performance gains are less pronounced. Notably, *FT* filtering is also competitive. Given the computational expense of XLM-RoBERTa embeddings, FastText can be a promising alternative in resource-constrained setups.

### 4.2.2 Threshold Selection

In Section 4.2.1, we base our model selection on experiments that retain top 10% of the data for high-resource languages. *But is this the optimal threshold?* Following the methodology of Li et al. [2024b], we analyze the impact of varying filter strengths on performance for Chinese, German, and French, using our *MLP* and *FT* filtering methods. The results are summarized in Table 2, with a comprehensive analysis, including results for *CS*, provided in Appendix C.2 (Table 14). Consistent with their findings, we observe that retaining top 10% of the data is a competitive threshold, particularly for approaches using the $MKC^+$ dataset. Interestingly, approaches using *MKC* perform better with higher retention. In Appendix C.2, we investigate how some filters' bias toward shorter documents affects threshold selection, though our analysis indicates multiple factors contribute to optimal threshold determination.

Table 2: Benchmark performance comparison: Average rank between FineWeb-2 baseline and our proposed filtering methods (*FT*, *MLP*) trained on $MKC^+$ or *MKC*, retaining top 10%, 15% or 20% of documents. The average rank is computed across FineTasks for 1B-parameter models evaluated on Chinese, German and French after 70B and 119B tokens.

| Approach | Threshold | Average Rank |
|----------|-----------|--------------|
| *MLP MKC$^+$* | 10% | 8.85 |
| *MLP MKC$^+$* | 15% | 9.44 |
| *MLP MKC* | 20% | 11.37 |
| *MLP MKC* | 15% | 11.70 |
| *MLP MKC* | 10% | 11.95 |
| *MLP MKC$^+$* | 20% | 11.97 |
| *FT MKC$^+$* | 10% | 13.92 |
| *FT MKC* | 15% | 14.62 |
| *FT MKC* | 10% | 14.74 |
| *FT MKC* | 20% | 15.62 |
| *FT MKC$^+$* | 15% | 16.27 |
| *FT MKC$^+$* | 20% | 16.51 |
| Baseline | – | 18.55 |

### 4.2.3 Training Data Analysis

The experiments in Sections 4.2.1 and 4.2.2 are based on the training datasets *MKC* and *MKC+*. *But is the diversity introduced by combining various base datasets truly necessary?* We evaluate the impact of each base dataset individually and compare it to the combined *MKC+* dataset. For this ablation study, we use our best filtering method (*MLP* with a top 10% retention) and train the models on 30B tokens. This token count is chosen to match the size of the smallest filtered dataset, ensuring consistency across comparisons.

The results, presented in Table 3, show that despite the absence of a quality guarantee for all samples in the *Aya Collection*, this dataset yields strong performance, making our approach applicable for various languages. Overall, we observe that the diversity resulting from combining all individual training datasets gives the best results.

Interestingly, models trained exclusively on *Include-Base-44* and *OpenAssistant-2* perform worse overall than the baseline. This may reflect dataset characteristics—*Include-Base-44* is small and domain-specific, containing mostly driving license exam questions in its German subset. *OpenAssistant-2* includes a limited number of samples, with fewer than 2K positive samples per training set, which likely negatively impacts classifier performance. In Appendix C.3, we reexamine how document length bias relates to model performance, confirming our Section 4.2.2 finding that performance depends on factors beyond document length. In Appendix C.4, we further verify our filtering approach preserves sufficient dataset diversity.

Table 3: Benchmark performance comparison: Average rank between FineWeb-2 baseline and *MLP* filtering trained on either full *MKC+* or its individual components, retaining top 10% for Chinese, German, and French, 56% for Arabic, and 65% for Danish. The average rank is computed across FineTasks for 1B-parameter models trained on 30B tokens per language.

| Dataset | Average Rank |
|---|---|
| *MKC+* | 2.52 |
| *Aya Collection* | 2.91 |
| *Aya Dataset* | 3.17 |
| *MMLU* | 3.57 |
| Baseline | 4.09 |
| *OpenAssistant-2* | 4.53 |
| *Include-Base-44* | 5.42 |

### 4.2.4 Approach Validation on English

Previous experiments have shown strong performance of our *MLP MKC+* approach. *But do these results translate to English?* Table 4 presents the performance of *MLP MKC+* with 10% retention applied to the English FineWeb dataset Penedo et al. [2024a]. Our method is compared against FineWeb and baselines using model-based filtered datasets, including DCLM Li et al. [2024b] and FineWeb-Edu [Penedo et al., 2024a]. To save computational resources, we use the 6 most recent FineWeb and FineWeb-Edu dumps and the first partition of DCLM[8], which we denote with *. Each of these subsets contains more than 119B tokens, with FineWeb retaining this size even after applying our filtering retaining top 10% of the documents.

Table 4: English benchmark performance: Our *MLP MKC+* approach (top 10% documents) compared to FineWeb, DCLM, and FineWeb-Edu baselines. The average rank is computed across SmolLM tasks using 1B-parameter models trained on 119B tokens.

| Dataset | Ours | DCLM* | FW-Edu* | FW* |
|---|---|---|---|---|
| Average Rank | 1.8333 | 2.3889 | 2.4444 | 3.3333 |
| ARC (Challenge) | 0.3550 | 0.3530 | **0.3850** | 0.3010 |
| ARC (Easy) | 0.6670 | 0.6470 | **0.6970** | 0.5880 |
| CommonsenseQA | 0.3870 | **0.4100** | 0.3770 | 0.3850 |
| HellaSwag | **0.6040** | 0.5960 | 0.5700 | 0.5930 |
| MMLU | 0.3400 | 0.3160 | **0.3470** | 0.3030 |
| OpenBookQA | 0.3860 | 0.3840 | **0.4180** | 0.3560 |
| PIQA | 0.7510 | 0.7510 | 0.7410 | **0.7620** |
| WinoGrande | **0.5720** | 0.5610 | 0.5660 | 0.5550 |
| TriviaQA | 0.0820 | **0.1240** | 0.0320 | 0.0370 |

While each approach demonstrates strengths in different benchmarks, as seen from Table 4 and Figure 2, the overall average rank results indicate that our method outperforms all other baselines.

### 4.2.5 Data Contamination Analysis

Our LLMs are never trained on benchmark datasets. *But is the strong performance observed in the previous sections primarily due to an increased ratio of data contamination?* To ensure the validity of our approach, we conduct decontamination experiments, as web crawl data may include evaluation benchmark tasks. While Li et al. [2024b] addressed similar concerns, our approach follows the

---

[8]huggingface.co/datasets/mlfoundations/dclm-baseline-1.0-parquet

methodology of Brown et al. [2020]. Specifically, we perform 13-gram decontamination of the LLM training data separately for English and French evaluation benchmarks. However, unlike the original approach, we remove the entire document if it is flagged as contaminated, using the implementation provided in DataTrove [Penedo et al., 2024b].

Tables 5 and 6 present the results of decontamination experiments for English and French, respectively. We used the following experimental setup (removed document contamination rates): baseline FineWeb English (0.16%), *MLP MKC$^+$* English with 10% retention (0.19%), baseline FineWeb-2 French (0.14%), and *MLP MKC$^+$* French with 10% retention (0.14%). All models were trained on 119B tokens. Additionally, we compare the results against equivalent training runs without decontamination to further analyze its impact. For an example of a contaminated sample, see Appendix G.

For English models, decontamination slightly reduces performance both for our approach and baseline FineWeb data. Even after decontamination, our approach still outperforms the baseline trained on non-decontaminated data. For French models, our approach performs similarly on decontaminated and non-decontaminated data, both outperforming baseline FineWeb-2. Interestingly, decontaminated baseline data yields better results than its non-decontaminated counterpart.

Table 5: English benchmark performance: Our *MLP MKC$^+$* approach (retaining top 10% documents) in both decontaminated ($D$) and non-decontaminated versions, compared to baseline FineWeb datasets with the same variants. The average rank is computed across SmolLM tasks for 1B-parameter models trained on 119B tokens.

| Dataset | Ours | Ours$_D$ | FW* | FW*$_D$ |
|---|---|---|---|---|
| Average Rank | 1.5000 | 2.1111 | 3.0556 | 3.3333 |
| ARC (Challenge) | **0.3550** | 0.3440 | 0.3010 | 0.2880 |
| ARC (Easy) | **0.6670** | 0.6520 | 0.5880 | 0.5700 |
| CommonsenseQA | 0.3870 | **0.4000** | 0.3850 | 0.3820 |
| HellaSwag | **0.6040** | **0.6040** | 0.5930 | 0.5890 |
| MMLU | **0.3400** | 0.3220 | 0.3030 | 0.3050 |
| OpenBookQA | **0.3860** | 0.3840 | 0.3560 | 0.3740 |
| PIQA | 0.7510 | 0.7590 | **0.7620** | 0.7600 |
| WinoGrande | **0.5720** | 0.5550 | 0.5550 | 0.5570 |
| TriviaQA | **0.0820** | 0.0380 | 0.0370 | 0.0250 |

Table 6: French Benchmark performance: Our *MLP MKC$^+$* approach (retaining top 10% of the documents) in both decontaminated ($D$) and non-decontaminated versions, compared to baseline FineWeb-2 datasets with the same variants. The average rank is computed across FineTasks for 1B-parameter models trained on 119B tokens.

| Dataset | Ours | Ours$_D$ | FW-2$_D$ | FW-2 |
|---|---|---|---|---|
| Average Rank | 2.0556 | 2.0556 | 2.7222 | 3.1667 |
| Belebele | 0.3533 | 0.3400 | **0.3778** | 0.3444 |
| HellaSwag | **0.5380** | 0.5350 | 0.5180 | 0.5180 |
| X-CSQA | 0.2740 | 0.2810 | 0.2730 | **0.2870** |
| XNLI 2.0 | **0.7400** | **0.7400** | 0.7070 | 0.7180 |
| FQuAD | 0.2803 | 0.2620 | **0.2890** | 0.2401 |
| MMLU | **0.2895** | 0.2875 | 0.2711 | 0.2706 |
| Mintaka | 0.0438 | **0.0797** | 0.0658 | 0.0712 |
| X-CODAH | 0.2667 | **0.2900** | 0.2800 | 0.2633 |
| ARC (Challenge) | **0.3180** | 0.3110 | 0.2880 | 0.2850 |

#### 4.2.6 Impact on Multilingual Training - Mitigating the Curse of Multilinguality

Although not our main focus, we found that our refined datasets boost the performance of multilingual models. We trained a multilingual 1B-parameter model on 595B tokens (119B per language), covering all five languages: Chinese, German, French, Arabic and Danish. We compared each language's results to its monolingual counterpart trained on 119B tokens. Training is performed once for our filtered data and once for original (unfiltered) FineWeb-2.

The results for French are presented in Table 7. Surprisingly, the *curse* of multilinguality [Chang et al., 2023] turns into a *benefit* for our quality filtered datasets: The multilingual model outperforms its monolingual counterpart, when both models have seen an equal amount of tokens of the language of interest. Meanwhile, for unfiltered training data, the multilingual LLM suffers from the *curse* as expected. The disappearance of the *curse* is consistent across all languages except Chinese. Detailed results for the other languages are provided in Appendix C.5.

Table 7: French benchmark performance: Multilingual LLMs ($M$) trained on FineWeb-2 or our *MLP MKC$^+$* refined dataset (retaining top 10% for Chinese, German and French, 56% for Arabic, 65% for Danish) with 595B tokens, compared to monolingual models trained on 119B tokens. The average rank is computed across FineTasks for 1B-parameter models.

| Dataset | Ours$_M$ | Ours | FW-2 | FW-2$_M$ |
|---|---|---|---|---|
| Average Rank | 1.8333 | 2.0556 | 3.0000 | 3.1111 |
| Belebele | **0.3667** | 0.3533 | 0.3444 | 0.3511 |
| HellaSwag | 0.5270 | **0.5380** | 0.5180 | 0.4970 |
| X-CSQA | 0.2740 | 0.2740 | **0.2870** | 0.2750 |
| XNLI 2.0 | **0.7660** | 0.7400 | 0.7180 | 0.7330 |
| FQuAD | **0.3212** | 0.2803 | 0.2401 | 0.2459 |
| MMLU | 0.2841 | **0.2895** | 0.2706 | 0.2735 |
| Mintaka | 0.0456 | 0.0438 | **0.0712** | 0.0579 |
| X-CODAH | **0.2900** | 0.2667 | 0.2633 | 0.2567 |
| ARC (Challenge) | 0.2970 | **0.3180** | 0.2850 | 0.2670 |

# 5   Conclusion

In this work, we developed a framework for model-based filtering of web-scale multilingual pretraining datasets, demonstrating consistent improvements on LLM benchmarks across a wide range of languages. Our Transformer embedding-based classifier, *MLP MKC$^+$*, outperforms state-of-the-art methods on both English and multilingual datasets, even when decontaminating the datasets or using them for training multilingual LLMs. While our FastText-based filtering approach performed well and shows promise in resource-constrained setups, *MLP MKC$^+$* consistently outperformed all other methods and can be easily scaled to other languages. These results provide strong empirical evidence supporting our expansion of the framework to 20 languages. We release the corresponding refined pretraining datasets and code, contributing to the advancement of multilingual language modeling.

## Acknowledgments and Disclosure of Funding

We thank Guilherme Penedo, Hynek Kydlíček, and Leandro von Werra for their help with FineWeb-2 data, and to Alex Hägele for providing feedback on the paper draft.

This work was supported as part of the Swiss AI Initiative by a grant from the Swiss National Supercomputing Centre (CSCS) under project ID a06 on Alps.

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

# A  Dataset Information

Based on the results of our experiments, we create the dataset, named ***FineWeb2-HQ***, by filtering all available FineWeb-2 data (version `2.0.1`) in 20 languages using the *MLP MKC$^+$* approach with 10% retention rate. The statistics of the resulting dataset are presented in Table 8. We release the dataset under the *Open Data Commons Attribution License (ODC-By) v1.0* license at huggingface.co/datasets/epfml/FineWeb2-HQ.

The main use case of our dataset is LLM pretraining, however, the dataset may also be used for other natural language processing tasks.

Table 8: Statistics (number of documents and disk size) of the dataset resulting from filtering FineWeb-2 using the *MLP MKC$^+$* approach with 10% retention rate in 20 languages.

| Language | Number of documents | Disk size |
|---|---|---|
| Russian | 55,220,956 | 1.2TB |
| Chinese | 54,211,986 | 784GB |
| German | 43,095,728 | 618GB |
| Spanish | 40,057,637 | 515GB |
| Japanese | 34,185,427 | 393GB |
| French | 32,248,772 | 483GB |
| Italian | 21,180,304 | 269GB |
| Portuguese | 18,135,468 | 222GB |
| Polish | 13,384,885 | 168GB |
| Dutch | 12,920,963 | 160GB |
| Indonesian | 8,911,149 | 125GB |
| Turkish | 8,578,808 | 100GB |
| Czech | 5,995,459 | 104GB |
| Arabic | 5,560,599 | 94GB |
| Persian | 5,107,187 | 69GB |
| Hungarian | 4,527,332 | 79GB |
| Swedish | 4,382,454 | 61GB |
| Greek | 4,346,440 | 84GB |
| Danish | 4,082,751 | 61GB |
| Vietnamese | 4,003,956 | 59GB |

# B  Limitations

A limitation of our work is that we perform experiments on relatively small 1B models with one seed per experiment. We use 1B models to balance the trade-off between the cost of pretraining and the measured signal from the experiments, as found in prior work [Penedo et al., 2024a,c, Li et al., 2024b]. Additionally, we compare our method to one multilingual baseline, FineWeb-2. However, since FineWeb-2 is the current state-of-the-art and due to our limited computational budget, we decided to allocate more compute towards understanding the mechanics of the data selection process, rather than confirming our results across previous datasets. Nevertheless, computational constraints prevented us from ablating every decision—such as our choice to use only the first 512 tokens for classification. We assume that if the first 512 tokens demonstrate good quality, the remainder of the document likely does as well. Given the strong performance achieved using the first 512 tokens, we prioritized this methodological simplicity. To facilitate further exploration of alternative selection strategies, we have made FineWeb2-embedded[9] available to the community, which contains embeddings for all 512-token chunks.

Although we develop our framework on languages from diverse language families, with different writing systems and with varying resource availability to find an approach that best generalizes for general web crawl text data across languages, classifier training datasets have no quality guarantees for other languages and may result in performance differences that are not visible in our experiments.

Since we focus on simple methods with broad availability and low computational cost, we discuss the computational cost difference between FastText and Transformer embeddings-based methods.

---

[9]huggingface.co/datasets/epfml/FineWeb2-embedded

While FastText classifiers are cheap to train and inference and can be efficiently run on CPU, Transformer-based methods require an initial computation of embeddings. To mitigate the higher cost of Transformer embeddings, we use a relatively small XLM-RoBERTa model and additionally release the dataset with precomputed embeddings[9]. The total cost for computing the embeddings is approximately 4K compute hours for the 20 languages.

We base our dataset on the FineWeb-2 dataset which conforms to Common Crawl robots.txt opt-outs (at crawl time), removes personally identifiable content, and offers a form for requesting data removal. Since ensuring privacy and fairness of our dataset is beyond the scope of this work, we make the dataset publicly available. This allows other researchers and the public to analyze potential biases, a critical task given that data curation is a political process that can introduce cultural and political impacts [Desai et al., 2024].

## C  Additional Results

### C.1  Model Selection - Per Language Results

For clarity, we present the individual benchmark results of the 1B-parameter model trained on 119B tokens for each language in the following tables: Table 9 for Chinese, Table 10 for French, Table 11 for German, Table 12 for Arabic, and Table 13 for Danish.

Table 9: Chinese Benchmark performance comparison: Average rank between FineWeb-2 baseline and our proposed filtering methods (*FT*, *MLP*, and *CS*) trained on $MKC^+$ or $MKC$, retaining top 10% of documents. The average rank is computed across FineTasks for 1B-parameter models evaluated after 119B tokens.

| Approach | MLP $MKC^+$ | MLP MKC | CS MKC | FT MKC | FT $MKC^+$ | Baseline | CS $MKC^+$ |
|---|---|---|---|---|---|---|---|
| Average Rank | 1.7333 | 2.4333 | 4.0667 | 4.0667 | 4.4667 | 5.2333 | 6.0000 |
| AGIEval | **0.2995** | 0.2948 | 0.2897 | 0.2919 | 0.2817 | 0.2853 | 0.2773 |
| Belebele | **0.3300** | 0.3233 | 0.3178 | 0.3133 | 0.3133 | 0.3056 | 0.3022 |
| C$^3$ | **0.4550** | 0.4480 | 0.4400 | 0.4500 | 0.4400 | 0.4400 | 0.4370 |
| C-Eval | **0.3095** | 0.3060 | 0.2760 | 0.2903 | 0.2906 | 0.2878 | 0.2805 |
| CMMLU | **0.3312** | 0.3259 | 0.3041 | 0.3043 | 0.3060 | 0.3009 | 0.2995 |
| CMRC 2018 | 0.2224 | 0.2125 | 0.1614 | **0.2251** | 0.2164 | 0.1949 | 0.1866 |
| HellaSwag | 0.3790 | **0.3800** | 0.3530 | 0.3680 | 0.3660 | 0.3510 | 0.3370 |
| M3Exam | **0.3319** | 0.3245 | 0.3084 | 0.3201 | 0.3245 | 0.3216 | 0.3245 |
| X-CODAH | 0.3033 | 0.3000 | **0.3233** | 0.3100 | 0.2900 | 0.2967 | 0.3067 |
| X-CSQA | **0.2740** | 0.2680 | 0.2690 | 0.2610 | 0.2520 | 0.2510 | 0.2650 |
| XCOPA | 0.6200 | **0.6400** | 0.6180 | 0.5740 | 0.5740 | 0.6000 | 0.5620 |
| OCNLI | 0.5470 | 0.5470 | 0.5340 | 0.5250 | **0.5600** | 0.5420 | 0.5060 |
| Chinese-SQuAD | 0.0929 | **0.1097** | 0.0865 | 0.0889 | 0.0850 | 0.0777 | 0.0585 |
| XStoryCloze | **0.5800** | 0.5630 | 0.5710 | 0.5560 | 0.5610 | 0.5580 | 0.5570 |
| XWINO | 0.6429 | 0.6528 | **0.6587** | 0.6131 | 0.5992 | 0.6429 | 0.6111 |

### C.2  Threshold Selection

**Complete Result.** To confirm that the *CS* filtering method is not competitive with *MLP* and *FT*, even when a higher percentage of documents is retained, we present the complete threshold selection results, including the *CS* method, in Table 14 in addition to the results shown in Section 4.2.2 (Table 2).

**Document Length Bias.** Motivated by the observed bias in certain approaches favoring the selection of shorter documents, as seen in Figure 3, Figure 4 and Table 15, we examine how this bias interacts with performance when retaining more documents. As demonstrated in Table 15, the *MLP MKC* approach shows a tendency to retain shorter documents, while achieving higher performance with an increased number of retained documents. In contrast, the *CS* and *FT* filtering methods present mixed results, suggesting that the optimal threshold selection may be influenced by additional factors.

Table 10: French Benchmark performance comparison: Average rank between FineWeb-2 baseline and our proposed filtering methods (*FT*, *MLP*, and *CS*) trained on *MKC⁺* or *MKC*, retaining top 10% of documents. The average rank is computed across FineTasks for 1B-parameter models evaluated after 119B tokens.

| Approach | *FT MKC⁺* | *MLP MKC⁺* | *MLP MKC* | *FT MKC* | *CS MKC* | *CS MKC⁺* | Baseline |
|---|---|---|---|---|---|---|---|
| Average Rank | 3.2222 | 3.5000 | 3.5556 | 3.7778 | 4.0000 | 4.6667 | 5.2778 |
| Belebele | 0.3378 | 0.3533 | **0.3678** | 0.3489 | 0.3444 | 0.3344 | 0.3444 |
| HellaSwag | **0.5380** | **0.5380** | 0.4990 | 0.5150 | 0.5280 | 0.5070 | 0.5180 |
| X-CSQA | 0.2820 | 0.2740 | 0.2730 | **0.2990** | 0.2850 | 0.2900 | 0.2870 |
| XNLI 2.0 | 0.7340 | 0.7400 | 0.7430 | 0.7230 | **0.7450** | 0.7330 | 0.7180 |
| FQuAD | 0.2597 | 0.2803 | **0.3032** | 0.2981 | 0.2411 | 0.2476 | 0.2401 |
| MMLU | 0.2896 | 0.2895 | **0.2925** | 0.2886 | 0.2806 | 0.2815 | 0.2706 |
| Mintaka | 0.0710 | 0.0438 | 0.0334 | 0.0670 | 0.0610 | **0.0976** | 0.0712 |
| X-CODAH | **0.3000** | 0.2667 | 0.2867 | 0.2767 | **0.3000** | 0.2800 | 0.2633 |
| ARC (Challenge) | 0.3120 | **0.3180** | 0.3090 | 0.3060 | 0.2950 | 0.2830 | 0.2850 |

Table 11: German Benchmark performance comparison: Average rank between FineWeb-2 baseline and our proposed filtering methods (*FT*, *MLP*, and *CS*) trained on *MKC⁺* or *MKC*, retaining top 10% of documents. The average rank is computed across FineTasks for 1B-parameter models evaluated after 119B tokens.

| Approach | *MLP MKC⁺* | *FT MKC⁺* | *FT MKC* | *CS MKC* | *MLP MKC* | *CS MKC⁺* | Baseline |
|---|---|---|---|---|---|---|---|
| Average Rank | 3.1250 | 3.1250 | 3.5000 | 3.7500 | 4.5000 | 4.7500 | 5.2500 |
| MMLU | **0.2940** | 0.2879 | 0.2926 | 0.2770 | 0.2905 | 0.2764 | 0.2718 |
| ARC (Challenge) | 0.2760 | 0.2850 | 0.2820 | **0.2880** | 0.2830 | 0.2640 | 0.2680 |
| Mintaka | 0.0580 | 0.0548 | 0.0735 | 0.0576 | 0.0494 | **0.0766** | 0.0498 |
| Belebele | **0.3611** | 0.3578 | 0.3544 | 0.3544 | 0.3567 | 0.3422 | 0.3544 |
| X-CODAH | 0.3367 | 0.3500 | 0.3300 | 0.3567 | 0.3400 | **0.3600** | 0.3467 |
| X-CSQA | 0.2978 | **0.3008** | 0.2877 | 0.2887 | 0.2857 | 0.2918 | 0.2787 |
| HellaSwag | 0.4640 | 0.4710 | **0.4870** | 0.4820 | 0.4540 | 0.4390 | 0.4470 |
| XNLI 2.0 | 0.6620 | 0.6530 | 0.6740 | 0.6440 | 0.6610 | 0.6520 | **0.6890** |

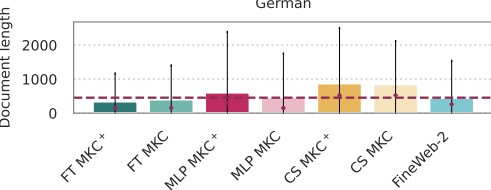

Figure 3: Document length comparison: Average length and standard deviation in FineWeb-2 before and after 10% retention filtering. Red horizontal line shows average document length, red dots indicate medians. Length measured by space-separated tokens.

### C.3 Training Data Analysis

We give details on the variation in the average length of documents retained by our model-based filtering method *MLP* for Chinese, French, Arabic, and Danish with different training datasets. The results are shown for German in Figure 5 and for all other languages in Figure 6.

### C.4 Replay of Original Data

We explore whether incorporating a small percentage of original raw data (replay) can help improve performance. We do this for our best FastText (*FT MKC⁺*) and Transformer approaches (*MLP MKC⁺*). Table 16 presents the results of experiments where 5% and 10% unfiltered data were mixed

Table 12: Arabic Benchmark performance comparison: Average rank between FineWeb-2 baseline and our proposed filtering methods (*FT*, *MLP*, and *CS*) trained on *MKC⁺* or *MKC*, retaining top 56% of documents. The average rank is computed across FineTasks for 1B-parameter models evaluated after 119B tokens.

| Approach | *MLP MKC⁺* | *MLP MKC* | *FT MKC⁺* | Baseline | *CS MKC⁺* | *CS MKC* | *FT MKC* |
|---|---|---|---|---|---|---|---|
| Average Rank | 2.7812 | 3.2500 | 3.6875 | 3.9688 | 3.9688 | 5.0312 | 5.3125 |
| EXAMS | 0.3537 | **0.3656** | 0.3552 | 0.3582 | 0.3443 | 0.3262 | 0.3346 |
| MMLU | 0.4007 | 0.3909 | **0.4023** | 0.3894 | 0.3912 | 0.3781 | 0.3885 |
| ARC (Easy) | **0.4330** | 0.4230 | 0.4210 | 0.4120 | 0.4020 | 0.3940 | 0.4080 |
| AlGhafa SciQ | 0.6915 | **0.7005** | 0.6965 | 0.6854 | 0.6724 | 0.6683 | 0.6804 |
| Belebele | 0.3456 | 0.3356 | 0.3322 | 0.3311 | 0.3356 | **0.3567** | 0.3233 |
| SOQAL | **0.7333** | 0.6867 | 0.7000 | 0.7200 | 0.7267 | 0.6867 | 0.7133 |
| MLQA | 0.2386 | **0.2402** | 0.1928 | 0.1901 | 0.2189 | 0.2154 | 0.1793 |
| TyDi QA | **0.1547** | 0.1476 | 0.1230 | 0.1441 | 0.1223 | 0.1097 | 0.1182 |
| AlGhafa RACE | 0.3720 | **0.3740** | 0.3640 | 0.3710 | 0.3590 | 0.3660 | 0.3730 |
| ARCD | **0.3638** | 0.3505 | 0.3235 | 0.3354 | 0.3358 | 0.3432 | 0.3043 |
| X-CODAH | 0.2600 | 0.2533 | 0.2567 | **0.2633** | **0.2633** | 0.2500 | 0.2600 |
| AlGhafa PIQA | 0.6360 | 0.6320 | **0.6400** | 0.6240 | 0.6320 | 0.6320 | 0.6370 |
| X-CSQA | 0.2740 | 0.2810 | 0.2770 | **0.2900** | 0.2880 | 0.2720 | 0.2770 |
| XNLI 2.0 | 0.6570 | 0.6910 | 0.6990 | **0.7010** | 0.6910 | 0.6900 | 0.6770 |
| HellaSwag | 0.4270 | 0.4220 | 0.4280 | 0.4250 | 0.4260 | **0.4320** | 0.4150 |
| XStoryCloze | 0.6150 | 0.6100 | 0.6100 | 0.6070 | 0.6130 | **0.6180** | 0.5930 |

Table 13: Danish Benchmark performance comparison: Average rank between FineWeb-2 baseline and our proposed filtering methods (*FT*, *MLP*, and *CS*) trained on *MKC⁺* or *MKC*, retaining top 65% of documents. The average rank is computed across FineTasks for 1B-parameter models evaluated after 119B tokens.

| Approach | *CS MKC⁺* | *MLP MKC⁺* | *FT MKC⁺* | Baseline |
|---|---|---|---|---|
| Average Rank | 1.0000 | 2.5000 | 3.1667 | 3.3333 |
| ARC (Challenge) | **0.2820** | 0.2650 | 0.2730 | 0.2560 |
| HellaSwag | **0.4950** | 0.4850 | 0.4750 | 0.4750 |
| Belebele | **0.3333** | 0.3289 | 0.3189 | 0.3289 |

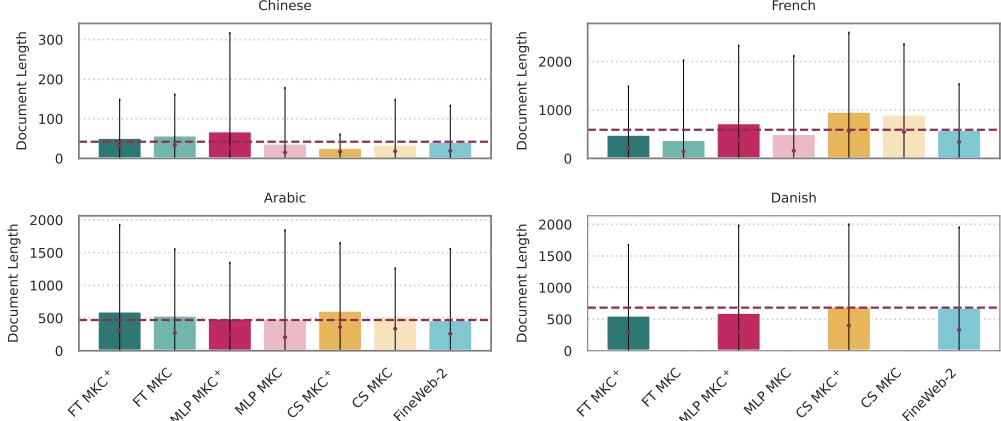

Figure 4: Document length comparison: Average length and standard deviation in FineWeb-2 before and after 10% retention filtering. Red horizontal line shows average document length, red dots indicate medians. Length measured by space-separated tokens.

into the training dataset, alongside results from training without any replay. Although, the *FT MKC⁺* filters shows mixed signal, our *MLP MKC⁺* approach clearly demonstrates that replay does not

Table 14: Benchmark performance comparison: Average rank between FineWeb-2 baseline and our proposed filtering methods (*FT*, *MLP*) trained on *MKC$^+$* or *MKC*, retaining top 10%, 15% or 20% of documents. The average rank is computed across FineTasks for 1B-parameter models evaluated on Chinese, German and French after 70B and 119B tokens.

| Approach | Threshold | Average Rank |
|---|---|---|
| *MLP MKC$^+$* | 10% | 11.73 |
| *MLP MKC$^+$* | 15% | 12.13 |
| *MLP MKC* | 20% | 15.07 |
| *MLP MKC* | 15% | 15.09 |
| *MLP MKC$^+$* | 20% | 15.40 |
| *MLP MKC* | 10% | 16.09 |
| *FT MKC$^+$* | 10% | 18.61 |
| *CS MKC* | 15% | 19.02 |
| *CS MKC* | 20% | 19.24 |
| *FT MKC* | 15% | 19.84 |
| *FT MKC* | 10% | 20.02 |
| *CS MKC* | 10% | 20.67 |
| *FT MKC* | 20% | 20.80 |
| *FT MKC$^+$* | 15% | 22.05 |
| *FT MKC$^+$* | 20% | 22.52 |
| *CS MKC$^+$* | 15% | 24.66 |
| *CS MKC$^+$* | 20% | 25.08 |
| Baseline | – | 25.54 |
| *CS MKC$^+$* | 10% | 26.94 |

Table 15: Token retention comparison: Counts in FineWeb-2 before and after filtering using our approach with 10% document retention for Chinese, French and German, 56% for Arabic, and 65% for Danish. Token counts represent tokenized dataset sizes using the multilingual Mistral v3 (Tekken) tokenizer [Mistral AI, 2024].

| Approach | Chinese | French | German | Arabic | Danish |
|---|---|---|---|---|---|
| *MLP MKC$^+$* | 150B (9%) | 89B (12%) | 119B (12%) | 78B (61%) | 71B (66%) |
| *MLP MKC* | 105B (7%) | 72B (10%) | 87B (9%) | 75B (59%) | – |
| *FT MKC$^+$* | 221B (14%) | 70B (10%) | 63B (6% ) | 77B (61%) | 70B (65%) |
| *FT MKC* | 190B (12%) | 43B (6%) | 65B (7%) | 80B (63%) | – |
| *CS MKC$^+$* | 170B (11%) | 126B (17%) | 166B (17%) | 82B (65%) | 77B (71%) |
| *CS MKC* | 161B (10%) | 132B (18%) | 172B (18%) | 83B (65%) | – |
| Baseline | 1597B | 730B | 973B | 127B | 108B |

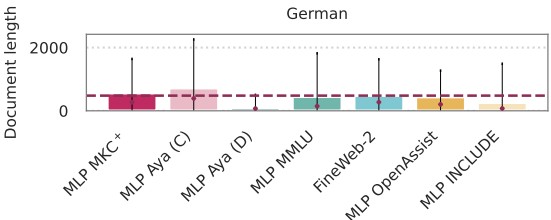

Figure 5: Document length comparison: Average length and standard deviation in FineWeb-2 before and after filtering using *MLP* method with 10% retention on different training datasets. Red horizontal line shows average document length, red dots indicate medians. Length measured by space-separated tokens.

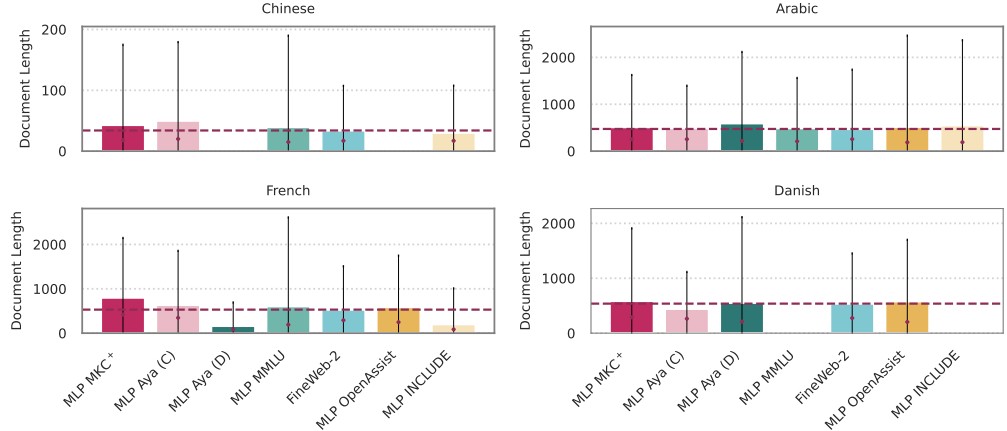

Figure 6: Document length comparison: Average length and standard deviation in FineWeb-2 before and after filtering using *MLP* method with 10% retention for Chinese and French, 56% for Arabic and 65% for Danish on different training datasets. Red horizontal line shows average document length, red dots indicate medians. Length measured by space-separated tokens.

improve performance, indicating the data selection already retains enough diversity. In cases of less diverse datasets, replay was shown to offer benefits [Bethune et al., 2025, Chen et al., 2023].

Table 16: Benchmark performance comparison: Average rank of our *MLP MKC*$^+$ and *FT MKC*$^+$ approaches with 10% document retention, mixed with 0%, 5%, or 10% of original FineWeb-2 dataset. The average rank is computed across FineTasks for 1B-parameter models evaluated on Chinese, German and French after 70B and 119B tokens.

| Approach | Mixture Rate | Average Rank |
|---|---|---|
| *MLP MKC*$^+$ | 5% | 5.09 |
| *MLP MKC*$^+$ | 0% | 5.16 |
| *MLP MKC*$^+$ | 10% | 5.40 |
| *FT MKC*$^+$ | 10% | 7.17 |
| *FT MKC*$^+$ | 0% | 7.51 |
| *FT MKC*$^+$ | 5% | 8.66 |

## C.5 Impact on multilingual model training

This section presents the results of our *MLP MKC*$^+$ approach on multilingual model training for Chinese (Table 17), Arabic (Table 18), German (Table 19), and Danish (Table 20), in addition to the results for French discussed in Section 4.2.6.

Table 17: Chinese benchmark performance: Multilingual LLMs ($M$) trained on FineWeb-2 or our *MLP MKC$^+$* refined dataset (retaining top 10% for Chinese, German and French, 56% for Arabic, 65% for Danish) with 595B tokens, compared to monolingual models trained on 119B tokens. The average rank is computed across FineTasks for 1B-parameter models.

| Dataset | Ours | Ours$_M$ | FW-2$_M$ | FW-2 |
|---|---|---|---|---|
| Average Rank | 1.5667 | 2.1667 | 2.9000 | 3.3667 |
| AGIEval | **0.2995** | 0.2863 | 0.2894 | 0.2853 |
| Belebele | 0.3300 | **0.3456** | 0.3189 | 0.3056 |
| C$^3$ | **0.4550** | 0.4520 | 0.4480 | 0.4400 |
| C-Eval | **0.3095** | 0.2848 | 0.2683 | 0.2878 |
| CMMLU | **0.3312** | 0.3064 | 0.2967 | 0.3009 |
| CMRC 2018 | 0.2224 | **0.2689** | 0.2090 | 0.1949 |
| HellaSwag | **0.3790** | 0.3740 | 0.3740 | 0.3510 |
| M3Exam | **0.3319** | 0.3040 | 0.3304 | 0.3216 |
| X-CODAH | 0.3033 | **0.3067** | 0.2800 | 0.2967 |
| X-CSQA | 0.2740 | **0.2810** | 0.2780 | 0.2510 |
| XCOPA | **0.6200** | 0.6020 | 0.5860 | 0.6000 |
| OCNLI | **0.5470** | 0.5320 | 0.4910 | 0.5420 |
| Chinese-SQuAD | 0.0929 | **0.1304** | 0.1017 | 0.0777 |
| XStoryCloze | **0.5800** | 0.5760 | 0.5650 | 0.5580 |
| XWINO | 0.6429 | 0.6409 | **0.6468** | 0.6429 |

Table 18: Arabic benchmark performance: Multilingual LLMs ($M$) trained on FineWeb-2 or our *MLP MKC$^+$* refined dataset (retaining top 10% for Chinese, German and French, 56% for Arabic, 65% for Danish) with 595B tokens, compared to monolingual models trained on 119B tokens. The average rank is computed across FineTasks for 1B-parameter models.

| Dataset | Ours$_M$ | Ours | FW-2 | FW-2$_M$ |
|---|---|---|---|---|
| Average Rank | 1.9688 | 2.0000 | 2.7500 | 3.2812 |
| EXAMS | 0.3336 | 0.3537 | **0.3582** | 0.3076 |
| MMLU | 0.3828 | **0.4007** | 0.3894 | 0.3599 |
| ARC (Easy) | 0.4190 | **0.4330** | 0.4120 | 0.3760 |
| AlGhafa SciQ | 0.6764 | **0.6915** | 0.6854 | 0.6563 |
| Belebele | **0.3511** | 0.3456 | 0.3311 | 0.3344 |
| SOQAL | 0.7000 | **0.7333** | 0.7200 | 0.6533 |
| MLQA | 0.2208 | **0.2386** | 0.1901 | 0.2085 |
| TyDi QA | **0.1634** | 0.1547 | 0.1441 | 0.1429 |
| AlGhafa RACE | **0.3830** | 0.3720 | 0.3710 | 0.3770 |
| ARCD | 0.3377 | **0.3638** | 0.3354 | 0.2970 |
| X-CODAH | **0.2767** | 0.2600 | 0.2633 | **0.2767** |
| AlGhafa PIQA | 0.6170 | **0.6360** | 0.6240 | 0.6160 |
| X-CSQA | 0.2860 | 0.2740 | **0.2900** | 0.2660 |
| XNLI 2.0 | 0.7080 | 0.6570 | 0.7010 | **0.7340** |
| HellaSwag | **0.4390** | 0.4270 | 0.4250 | 0.4240 |
| XStoryCloze | **0.6370** | 0.6150 | 0.6070 | 0.6160 |

Table 19: German benchmark performance: Multilingual LLMs ($M$) trained on FineWeb-2 or our *MLP MKC$^+$* refined dataset (retaining top 10% for Chinese, German and French, 56% for Arabic, 65% for Danish) with 595B tokens, compared to monolingual models trained on 119B tokens. The average rank is computed across FineTasks for 1B-parameter models.

| Dataset | Ours$_M$ | Ours | FW-2 | FW-2$_M$ |
|---|---|---|---|---|
| Average Rank | 1.5000 | 2.1250 | 2.9375 | 3.4375 |
| MMLU | 0.2918 | **0.2940** | 0.2718 | 0.2691 |
| ARC (Challenge) | 0.2740 | **0.2760** | 0.2680 | 0.2640 |
| Mintaka | **0.0821** | 0.0580 | 0.0498 | 0.0660 |
| Belebele | **0.3956** | 0.3611 | 0.3544 | 0.3633 |
| X-CODAH | **0.3500** | 0.3367 | 0.3467 | 0.3167 |
| X-CSQA | **0.3048** | 0.2978 | 0.2787 | 0.2787 |
| HellaSwag | **0.4690** | 0.4640 | 0.4470 | 0.4430 |
| XNLI 2.0 | 0.6420 | 0.6620 | **0.6890** | 0.6340 |

Table 20: Danish benchmark performance: Multilingual LLMs ($M$) trained on FineWeb-2 or our *MLP MKC$^+$* refined dataset (retaining top 10% for Chinese, German and French, 56% for Arabic, 65% for Danish) with 595B tokens, compared to monolingual models trained on 119B tokens. The average rank is computed across FineTasks for 1B-parameter models.

| Dataset | Ours$_M$ | Ours | FW-2$_M$ | FW-2 |
|---|---|---|---|---|
| Average Rank | 1.6667 | 2.1667 | 3.0000 | 3.1667 |
| ARC (Challenge) | **0.2920** | 0.2650 | 0.2600 | 0.2560 |
| HellaSwag | 0.4710 | **0.4850** | 0.4560 | 0.4750 |
| Belebele | **0.3700** | 0.3289 | 0.3311 | 0.3289 |

# D List of evaluation benchmarks and metrics

We provide a detailed overview of the evaluation benchmarks used to assess our models' performance, along with their respective evaluation metrics in Table 21. For non-English tasks and English MMLU, we use the *cloze* multiple-choice prompt, which allows the model to directly predict each option instead of using the standard prompt format with A/B/C/D letter prefixes as targets. This approach was chosen because it has been shown to serve as a more reliable performance indicator earlier in training [Kydlíček et al., 2024]. We evaluate the models in a 0-shot setting.

Table 21: List of Evaluation Benchmarks and Metrics used in our setup for Chinese, French, German, Arabic, Danish, and English.

| Benchmark | Chinese | French | German | Arabic | Danish | English | Evaluation metric |
|---|---|---|---|---|---|---|---|
| AGIEval [Zhong et al., 2023] | ✓ | | | | | | Normalized accuracy |
| AlGhafa ARC [Almazrouei et al., 2023] | | | | ✓ | | | Normalized accuracy |
| AlGhafa PIQA [Almazrouei et al., 2023] | | | | ✓ | | | Normalized accuracy |
| AlGhafa RACE [Almazrouei et al., 2023] | | | | ✓ | | | Normalized accuracy |
| AlGhafa SciQ [Almazrouei et al., 2023] | | | | ✓ | | | Normalized accuracy |
| ArabicMMLU [Koto et al., 2024] | | | | ✓ | | | Normalized accuracy |
| ARC [Clark et al., 2018] | | | | | | ✓ | Normalized accuracy |
| ARCD [Mozannar et al., 2019] | | | | ✓ | | | F1 score |
| Belebele [Bandarkar et al., 2024] | ✓ | ✓ | ✓ | ✓ | ✓ | | Normalized accuracy |
| C$^3$ [Sun et al., 2020] | ✓ | | | | | | Normalized accuracy |
| C-Eval [Huang et al., 2023] | ✓ | | | | | | Normalized accuracy |
| Chinese-SQuAD [Pluto-Junzeng, 2019] | ✓ | | | | | | F1 score |
| CMMLU [Li et al., 2024a] | ✓ | | | | | | Normalized accuracy |
| CMRC 2018 [Cui et al., 2019] | ✓ | | | | | | F1 score |
| CommonsenseQA [Talmor et al., 2019] | | | | | | ✓ | Normalized accuracy |
| EXAMS [Hardalov et al., 2020] | | | | ✓ | | | Normalized accuracy |
| FQuAD [d'Hoffschmidt et al., 2020] | | ✓ | | | | | F1 score |
| HellaSwag [Zellers et al., 2019] | | | | | | ✓ | Normalized accuracy |
| M3Exam [Zhang et al., 2023] | ✓ | | | | | | Normalized accuracy |
| Meta MMLU [Llama Team, 2024] | | ✓ | ✓ | | | | Normalized accuracy |
| Mintaka [Sen et al., 2022] | | ✓ | ✓ | | | | F1 score |
| MLMM ARC [Lai et al., 2023] | | ✓ | ✓ | | ✓ | | Normalized accuracy |
| MLMM HellaSwag [Lai et al., 2023] | ✓ | ✓ | ✓ | ✓ | ✓ | | Normalized accuracy |
| MLQA [Lewis et al., 2020] | | | | ✓ | | | F1 score |
| MMLU [Hendrycks et al., 2020] | | | | | | ✓ | Normalized accuracy |
| OCNLI [Hu et al., 2020] | ✓ | | | | | | Normalized accuracy |
| OpenBookQA [Mihaylov et al., 2018] | | | | | | ✓ | Normalized accuracy |
| PIQA [Bisk et al., 2019] | | | | | | ✓ | Normalized accuracy |
| SOQAL [Mozannar et al., 2019] | | | | ✓ | | | Normalized accuracy |
| TriviaQA [Joshi et al., 2017] | | | | | | ✓ | Quasi-exact match |
| TyDi QA [Clark et al., 2020] | | | | ✓ | | | F1 score |
| WinoGrande [Sakaguchi et al., 2019] | | | | | | ✓ | Normalized accuracy |
| X-CODAH [Lin et al., 2021a] | ✓ | ✓ | ✓ | ✓ | | | Normalized accuracy |
| XCOPA [Ponti et al., 2020] | ✓ | | | | | | Normalized accuracy |
| X-CSQA [Lin et al., 2021a] | ✓ | ✓ | ✓ | ✓ | | | Normalized accuracy |
| XNLI 2.0 [Upadhyay and Upadhya, 2023] | | ✓ | ✓ | ✓ | | | Normalized accuracy |
| XStoryCloze [Lin et al., 2021b] | ✓ | | | ✓ | | | Normalized accuracy |
| XWINO [Tikhonov and Ryabinin, 2021] | ✓ | | | | | | Normalized accuracy |

# E Average Rank Computation

Analogous to the method in FineTasks [Kydlíček et al., 2024], we compute the average rank for our ablations as follows:

1. We train a model for each parameter configuration we want to ablate on.
2. We evaluate each model on all the selected benchmarks.
3. We compute the rank of each model (individual experiment) with regard to each benchmark and language.
4. We compute the average rank for each model across all benchmarks and languages.

# F  FineWeb documents in different scoring approaches

To illustrate the types of documents each classifier scores highly or poorly, we present the highest- and lowest-scoring FineWeb examples for each of our classifier approaches (*FT MKC$^+$*, *MLP MKC$^+$*, *CS MKC$^+$*). These examples were selected from the randomly chosen FineWeb test dataset (10K samples) used to validate the training of our model-based classifiers.

## F.1  FastText Classifier (FT)

> **Highest score:**
>
> hi. i couldn't solve my problem because it has two conditional logical propositions. the problem is:can anyone help me about this, thanks =)we're expected to know that: . is equivalent tofind a logically equivalent proposition for:by first writing its contrapositive, and then applying demorgan's lawand the equality forthey were trying to be helpful by outlining the steps we should follow,. . but i think they made it more confusing.i don't see the purpose of using the contrapositive here.. . i wouldn't have done it that way.besides, the statement is a tautology . . .which gives us: .and this is a tautology: "a thing implies itself" ... which is always true.i don't know of any "logically equivalent proposition" we can write . . .

> **Lowest score:**
>
> |starts||23 sep 2016 (fri) (one day only)|want to travel soon but donâĂŹt wish to fork out a
> fortune for flights? check out todayâĂŹs promotion from jetstar featuring promo fares
> fr $35 all−in valid for travel period commencing 12 october 2016donâĂŹt miss out!
> all−in frenzy fares to hong kong, penang and more from $35.sale ends 23 sep, 11pm!|
> travelling||price||travel period||find flight||penang||$35^|| [...]

## F.2  Multi-Layer Perceptron (MLP)

> **Highest score:**
>
> Naqhadeh County is a county in West Azerbaijan Province in Iran. The capital of the county is Naqadeh. At the 2006 census, the county's population was 117,831, in 27,937 families. The county is subdivided into two districts: the Central District and Mohammadyar District. The county has two cities: Naqadeh and Mohammadyar.

> **Lowest score:**
>
> Custom Wedding Gifts
> Personalized photo frames, albums & keepsakes. Heirloom quality!
> Custom Engraved Journals
> Handmade in Florence Italy. Dozens of sizes and paper styles!
> Awesome Leather Journals
> Personalized, Customizable, Artisan made in Santa Fe, NM.
> Ink Rendering from Photos
> 100% Hand painted with unique style by pro artists. From $49.

### F.3 Cosine Similarity (CS)

> **Highest score:**
>
> When you are renting a 5, 10, 15, 20, 30 or 40 yard dumpster, you want a company you can trust with prices that make you smile. Give us a call today and see the difference we can make in your next construction or clean out project.
> Simply give us a call and we will help you figure out your dumpster rental needs.
> Our dumpsters usually go out same-day or next-day depending on when you call.
> We provide top-notch service, while going easy on your bottom line. What more could you ask for?
> Our trained operators are here to give you a fast and hassle-free experience from start to finish.[...]

> **Lowest score:**
>
> Cooperative flat 206/J
> - Cooperative flat 201/J - Sold
> 2(1)+kitchenette, 50,1 m2Cooperative flat 202/J - Sold
> 2(1)+kitchenette, 44,9 m2Cooperative flat 203/J - Sold
> 2(1)+kitchenette, 50,6 m2Cooperative flat 204/J - Sold
> 1+kitchenette, 27,1 m2Cooperative flat 205/J - Sold
> 2(1)+kitchenette, 50,1 m2Cooperative flat 206/J - On sale
> 3+kitchenette 86,7 m2[...]

## G   Example of a contaminated document

We present an example of a FineWeb document that was removed during our decontamination pipeline.

> **MMLU contaminated document (matched 13-gram in bold):**
>
> Here is our diagram of the Preamble to the Constitution of the United States. It is based on our understanding of the use of "in order to" as a subordinating conjunction that introduces a series of infinitival clauses (without subjects) that, in turn, modify the compound verbs "do ordain" and "establish."
> See A Grammar of Contemporary English by Randolph Quirk, Sidney Greenbaum, Geoffrey Leech, and Jan Svartvik. Longman Group: London. 1978. p. 753.
> We the People of the United States, in Order to form a more perfect Union, establish Justice, insure domestic Tranquility, **provide for the common defence, promote the general Welfare, and secure the Blessings** of Liberty to ourselves and our Posterity, do ordain and establish this Constitution for the United States of America.
> If you have alternative rendering for this sentence, we would be happy to hear of it. Use the e-mail icon to the left.

## H   License Information

### H.1   Dataset Licenses

We use the following pretraining datasets:

- FineWeb-2 (ODC-By license)
- FineWeb (ODC-By license)
- FineWeb-Edu (ODC-By license)
- DCLM (CC-BY 4.0)

We use the following classifier training datasets:

- Aya Collection (Apache 2.0 license)
- Aya Dataset (Apache 2.0 license)
- Translated multilingual MMLU [OpenAI, 2024] (MIT license)
- OpenAssistant-2 (Apache 2.0 license)
- Include-Base-44 (Apache 2.0 license)

## H.2 Code Licenses

We use the following open source code:

- Nanotron (Apache 2.0 license)
- Datatrove (Apache 2.0 license)
- Lighteval (MIT license)
- FastText (MIT license)

## H.3 Model Licenses

We use the following models:

- Mistral v3 (Tekken) (Apache 2.0 license)
- XLM-RoBERTa (MIT license)

