# OpenReview forum: "Enhancing Multilingual LLM Pretraining with Model-Based Data Selection"
_NeurIPS.cc/2025/Datasets_and_Benchmarks_Track — NeurIPS 2025 Datasets and Benchmarks Track poster_

### Official Review · Reviewer_PkEL · 2025-06-24

**Rating:** 4
**Confidence:** 3

**Summary:**

In this paper, the authors proposed a new data curation framework for curating multilingual datasets that are used for pre-training LLMs. Relevant studies are generally ignored in prior studies. To accomplish this goal, the authors employ a series of data curation techniques, including collecting representative samples, pre-processing samples, filtering invalid samples using models and etc. Extensive experiments can demonstrate the effectiveness of the proposed solutions and the authors also released a refined training dataset to the community.

**Additional Feedback:**

Not applicable

**Dataset Code Accessibility:**

Yes

**Dataset Code Comments:**

Not applicable

**Ethical Considerations:**

No, there are no or only very minor ethics concerns

**Final Justification:**

I have taken a pass over the authors' feedback and other reviewers' comments. Overall, I think the author's responses are reasonable and I would love to maintain my current score.

**Limitations Weaknesses:**

+ Some technical details may need more justification. For example, in Section 3.1, the authors point out that only the first 512 tokens are collected for each document and assume that those tokens are representative. However, intuitively speaking, for a large document, the first 512 tokens are obviously insufficient to represent this document. I guess for this scenario, random sampling within a large document may make more sense?
+ It seems that the authors did not compare the proposed data curation framework against other data curation methods, in particular, the ones mentioned in the related work section (e.g., the ones based on heuristics)
+ The authors only performed experiments with 1B llama model. It would be beneficial to repeat the experiments with varied model types and model sizes to extensively evaluate the effectiveness of the proposed curation method.

**Strengths Contributions:**

+ The overall data curation pipeline for constructing pre-trained multilingual datasets is comprehensive and sound.
+ The authors performed extensive empirical studies to evaluate the effectiveness of the proposed framework
+ The overall paper writing and presentation are clear and easy to follow.

---

> ### Author Rebuttal · Authors · 2025-07-30
>
> Dear reviewer PkEL,
>
> We thank you for your positive review and acknowledgement of our work.
>
> **W1:** We agree with the reviewer that other strategies for selecting 512 tokens could be explored. Our assumption is that if the first 512 tokens have good quality, the rest of the document does as well. Given the strong performance achieved using the first 512 tokens, we prioritized this methodological simplicity. We will make this decision clearer in the manuscript. To facilitate further exploration of alternative selection strategies, we have also made FineWeb2-embedded available to the community, which contains embeddings for all 512 token chunks.
>
> **W2:** We would like to point out that our baseline FineWeb-2 achieves state-of-the-art performance on heuristic-based filtering, which we demonstrate can be improved upon with our filtering approach. We do not view our method as a replacement, but rather as a valuable addition.
>
> **W3:** As mentioned in Appendix Section B, we use 1B models to balance the trade-off between the cost of pretraining and the measured signal from the experiments, as found in prior work [1][2].
>
> While recent LLMs are predominantly decoder-only transformer architecture (LLama [3], Qwen [4], Gemini [5]), we acknowledge that they may differ in minor adaptations [3]. However, comprehensive evaluation across all model families would require significant computational resources and is beyond the scope of this dataset contribution. Our publicly released dataset enables future work to explore model-specific variations.
>
> [1] DataComp-LM: In search of the next generation of training sets for language models, Li et al.
>
> [2] The FineWeb Datasets: Decanting the Web for the Finest Text Data at Scale, Penedo et al.
>
> [3] The Llama 3 Herd of Models, Llama Team, AI @ Meta
>
> [4] https://qwen.readthedocs.io/en/stable/getting_started/concepts.html
>
> [5] Gemini: A Family of Highly Capable Multimodal Models, Gemini Team, Google

---

### Official Review · Reviewer_dJ3W · 2025-07-02

**Rating:** 4
**Confidence:** 3

**Summary:**

Current rule or model-based data curation techniques focus mainly on English, but this study develops a multilingual dataset filtering framework to identify diverse, structured, and knowledge-rich data samples. The framework achieves baseline MMLU scores with only 15% of training tokens, improves performance across benchmarks and extends to 20 languages with a refined pre-training dataset released.

**Dataset Code Accessibility:**

Yes

**Dataset Code Comments:**

I can access the attached datasets.

**Ethical Considerations:**

No, there are no or only very minor ethics concerns

**Final Justification:**

Most of my concerns have been addressed during rebuttal period.
Regarding W3, I reviewed the case studies in Appendix F, which are sound. However, as the authors noted, we likely cannot draw general conclusions based on individual examples.
While this does not affect the soundness of the method, providing such analysis through a quantitative approach would enhance our understanding of the features of different selection methods.

**Limitations Weaknesses:**

1. Different model families can vary in their sensitivity to multilingual datasets, which raises the question of whether this extracted dataset can generalize to other model architectures (not the scale, I agree with authors' acknowledgement of scaling in the Appendix). To confirm the dataset's robustness and broad applicability, conducting ablation studies or sensitivity analyses would be beneficial.

2. The experimental results show that FastText-based Filtering (FT) is an effective classifier for high-quality text. The choice to use it as the primary method is justified by its proven effectiveness in prior studies. However, its specific advantages in multilingual contexts aren't fully clear, despite the intuitive benefits of multilingual embeddings. A broader comparison of classification methods, such as including decoder-based classifiers or incorporating model-based uncertainty metrics, would strengthen the claim that FT is the optimal choice.

3. An analysis of high- and low-scoring samples would help validate the soundness of the method. It would be particularly interesting to see which types of samples contribute to the performance improvement of FT over MLP and CS. Specifically, what features does the FT architecture capture more effectively to achieve these gains?

4. Why are representative samples considered more effective for training? One could argue that diversity or even noise in the data might enhance other aspects of model performance, such as robustness. The authors could strengthen their paper by including experiments that compare models trained on datasets from different scoring ranges to validate the superiority of representative samples.

**Strengths Contributions:**

1. The research background is well-organized and the paper is clearly written.
2. The study extends the scope of previous data curation work to multilingual settings and demonstrates the potential for mitigating the curse of multilinguality.
3. The released pre-training dataset is valuable for community to perform efficient multilingual pre-training.

---

> ### Author Rebuttal · Authors · 2025-07-30
>
> Dear reviewer dJ3W,
>
> We thank you for your positive review and acknowledgement of our work.
>
> **W1:** While recent LLMs are predominantly decoder-only transformer architecture (LLama [1], Qwen [2], Gemini [3]), we acknowledge that they may differ in minor adaptations [3]. However, comprehensive evaluation across all model families would require significant computational resources and is beyond the scope of this dataset contribution. Our publicly released dataset enables future work to explore model-specific variations.
>
> **W2:** We would like to emphasize that we do not claim the FT classifier is the optimal choice. Rather, we demonstrate that a single-hidden-layer neural network for binary classification using XML-RoBERTa document embeddings outperforms our other classification approaches. Since our focus is on text representation rather than text generation, we selected XML-RoBERTa, an encoder model, as our primary model for comparison.
>
> **W3:** Please refer to Appendix Section F, where we present the highest and lowest scoring samples from a randomly selected test set of 10K samples from FineWeb for each classifier. While we cannot make general claims about the quality of high-scoring samples based on individual examples, these samples reflect the overall performance of the three approaches. All classifiers assigned high scores to coherent text and low scores to noisy text. However, their highest-scoring samples differ notably: the MLP classifier selects an educational text about Naqhadeh County, FT selects a text about logical expressions (though the expression itself is missing), and the CS approach selects an advertisement.
>
> **W4:** While we thank the reviewer for suggesting this additional experiment, we note that selecting the highest quality samples is common practice [1][2]. We would like to draw the reviewers' attention to Sections 4.2.2, 4.2.3, and Appendix Section C.4, where **we evaluated our methods across different variations including noise and diversity considerations**, even though we did not train on different threshold ranges.
> - **Section 4.2.2** demonstrates that higher thresholds yield better performance for MLP MKC+ compared to lower thresholds, including the baseline of selecting samples from the complete score range.
> - **Section 4.2.3** indicates that diversity in the representative samples is important, as evidenced by the superior performance achieved with more diverse sets of representative samples.
> - **Appendix Section C.4** shows that replaying a small amount (5%) of original data with MLP MKC+ to increase diversity and include noise can provide a performance boost. However, we believe the overall strong performance of MLP MKC+ still justifies our filtering approach, particularly since original data can be added at any desired ratio.
>
> We believe these experiments demonstrate that **our approach exhibits strong performance across various setups**. While our publicly available code and datasets will enable further exploration in this direction, this experiment was beyond the scope of the current work.
>
>
>
> [1] The Llama 3 Herd of Models, Llama Team, AI @ Meta
>
> [2] https://qwen.readthedocs.io/en/stable/getting_started/concepts.html
>
> [3] Gemini: A Family of Highly Capable Multimodal Models, Gemini Team, Google

---

> > ### Comment · Reviewer_dJ3W · 2025-08-05
> >
> > Thank you for the clarifications. Most of my concerns have been addressed. Regarding W3, I reviewed the case studies in Appendix F, which are sound. However, as the authors noted, we likely cannot draw general conclusions based on individual examples. While this does not affect the soundness of the method, providing such analysis through a quantitative approach would enhance our understanding of the features of different selection methods.
> >
> > I will maintain a positive rating.

---

### Official Review · Reviewer_cncU · 2025-07-03

**Rating:** 5
**Confidence:** 4

**Summary:**

This paper proposes an approach for model-base data selection for constructing pretraining datasets for multi-lingual LLMs. They propose a similar approach to DCLM based on training a FastText classifier for detecting high-quality pages. Notably, they train a 1B-parameter Llama model using 70B and 119B tokens that batches MMLU score and improved multilingual performance. Particularly, they achieve more than 2% improvement on Chinese/German/French MMLU benchmarks compared to FineWeb-2.

The method mainly relies on training a FastText classifier on a representative diverse set of structured and knowledge-rich multilingual samples. For this purpose, they select 5 datasets based on two criteria. These datasets include both human data as well as automatically generated. In total there are ~514M samples in these datasets dominated by Aya Collection. The data is used to create two collections, MKC and MKC+, where MKC excludes Aya Collection. To train filtering models for each individual language, 80k random samples are selected from one of the collections together with 80k negative samples from FineWeb-2 pretraining dataset.

Three filtering methods are considered: 1) FastText-based and 2) Transformer embeddings with MLP and 3) Transformer embeddings with cosine similarity. For the first one, a FastText binary classifier is trained using 2-gram or 4-gram features. For second and third, a multilingual XLM-RoBERTa model is used to compute embeddings on the first 512 tokens of each document. Then either an MLP is trained on the embeddings or the cosine similarity to randomly sampled positive samples are used. Figure 1 suggests that MLP on MKC+ is overall preferred to a FastText classifier trained on the same set.

They apply their filtering to FastWeb-2 which is a multilingual public pretraining dataset. For English-only, they apply their method to FineWeb, FineWeb-Edu, and DCLM. The paper releases FineWeb2-HQ by filtering FineWeb-2 in 20 languages using their MLP MKC+ approach and targeting a 10% retention rate.

**Additional Feedback:**

Questions
- Does MKC+ have any balanced sampling to prevent Aya Collection from dominating the training? Aya Collection is ~2000x larger than other datasets combined.
- Section 4.2.1: What is the runtime efficiency of the recommended approach, transformer embedding with MLP classifier, compared with FastText classifier? It is noted in the appendix that for embedding computation the total compute time is 4k compute hours for 20 languages. Can authors elaborate as to how many CPUs/GPUs were used.
- Table 1: FT MKC+ is noticeably worse than MLP MKC+ according to the average rank metric. Why is it considered as “competitive” (Line 284)?
- Section 4.2.2: It seems that lower thresholds are generally better. Can one consider thresholds below 10%?

**Dataset Code Accessibility:**

Yes

**Dataset Code Comments:**

The released data is available on HuggingFace and can be loaded using HF API.

**Ethical Considerations:**

No, there are no or only very minor ethics concerns

**Final Justification:**

I thank authors for their response. I retain my positive rating. I had mainly requested additional information which was provided by the authors. I recommend also including the hardware specifications and clarifying whether CPU hours is per CPU core or CPU node.

**Limitations Weaknesses:**

The work is well polished with comprehensive analysis. One minor weakness is to include additional information about the computational cost of filtering with different models and the resource requirements. I also have a few questions listed under additional feedback.

**Strengths Contributions:**

- Multilingual LLM pretraining is an important problem with less attention paid by recent works exploring data filtering. This paper closes the gap.
- Section 4.2.1 shows the best performing filtering method, MLP MKC+, performs consistently well across languages.
- Section 4.2.3 shows the combination of datasets considered for positive samples in MKC+ are better than individual datasets.
- Section 4.2.4 shows that the proposed filtering by MLP MKC+ leads to improved results on English-only evaluations as well.
- Section 4.2.5 provides a comprehensive analysis of dataset contamination and its effect on performance.
- Section 4.2.6 shows that even the curse of multilinguality may not be affecting the proposed filtered dataset, that is the performance of training a multi-lingual model outperforms its monolingual counterpart at similar training tokens.

---

> ### Author Rebuttal · Authors · 2025-07-30
>
> Dear reviewer cncU,
>
> We thank you for your positive review and acknowledgement of our work.
>
> **W1:** Please refer to Q2.
>
> **Q1:** Thank you for pointing this out. We did not use balanced sampling. For small datasets, we used all available data. For the Aya collection, we randomly sampled up to 80K samples total, to ensure it is not dominating all others. We will clarify this in the manuscript.
>
> **Q2:** The computation of embeddings required approximately 4K GPU hours. We release the embeddings publicly so this computation does not have to be repeated. Data selection classifier training (i.e. FastText and MLP) can be completed within a few minutes on a CPU. Similarly, cosine similarity can be represented as matrix multiplication and parallelized on a CPU. Filtering is also performed on a CPU and can be easily parallelized. Overall, we spent around 60 CPU hours filtering German, Chinese, French, Arabic, and Danish for one filtering approach in the ablations. Thank you for bringing this to our attention. We will include these additional computational details for filtering together with the compute environment details in the manuscript.
>
> **Q3:** We consider FT MKC+ a competitive alternative based on Figure 2, which shows that FT MKC+ achieves performance very close to MLP MKC+ on MMLU for German, French, Arabic, and Danish. While FT MKC+ shows promise, its effectiveness should be evaluated on a case-by-case basis depending on the specific language and use case.
>
> **Q4:** This is a good point, we decided against running ablations with lower thresholds than 10%, as it already results in datasets with often less than 100BT. Please refer to Table 15 for exact token counts.

---

> > ### Comment · Reviewer_cncU · 2025-08-05
> >
> > I thank authors for their response. I retain my positive rating. Regarding computational cost, I recommend also including the hardware specifications and clarifying whether CPU hours is per CPU core or CPU node.

---

### Official Review · Reviewer_Eh93 · 2025-07-03

**Rating:** 4
**Confidence:** 4

**Summary:**

This paper studies the question of multilingual data curation. Specifically, it builds a filtered version of FineWeb-2 web crawl dataset with a few different model-based filters, including:

* FastText-based filter based on sparse n-gram features
* Embedding-based filters, including MLP (on top of the embedding) and cosine similarities of among groups of embeddings

Experiments build different versions of 1B-parameter Llama LLMs on the filtered datasets and evaluate performance on FineTasks dataset. The main takeaway from the results are that (1) The training dataset quality for the filtering model is important (2) MLP on average works better than group cosine similarity and fast-text based filters. The paper also studied other related questions to better understand the filtering effect, such as filtering threshold, data ablation, data contamination, etc.

**Additional Feedback:**

### Other Related Work

- Potentially relevant related work -- similar but specifically for LLM-based MT: https://www2.statmt.org/wmt24/pdf/2024.wmt-1.126.pdf
- For Section 4.2.5, you can consider citing/discussing (1) https://arxiv.org/abs/2303.03919 (2) https://arxiv.org/abs/2401.17377
- Contemporary work: https://huggingface.co/utter-project/EuroFilter-v1

### Misc. Questions/Feedback

* The title in L142 vs. 162 is very similar and it's not entirely obvious to me what's the difference between "Selection" vs. "Collection"
* L179 -- Was 80k for each language or for all the languages?
* L376 -- "Although not our main focus..." I'm confused here -- isn't improving multilingual model via data curation the focus of this paper?

**Dataset Code Accessibility:**

Yes

**Dataset Code Comments:**

I was able to access the data, see its first few rows in the data viewer, and there is a detailed dataset card attached to the data.

**Ethical Considerations:**

No, there are no or only very minor ethics concerns

**Final Justification:**

The authors have adequately addressed my comments and promised to make improvements on the current draft. All other reviewers seem to concur acceptance. I'm maintaining my score.

**Limitations Weaknesses:**

* As the authors rightfully pointed out, there has already been a lot of curated multilingual LLM datasets, such as madlad-400, CulturaX, etc. It is not entirely clear to me whether or why this data adds extra value to the existing available data -- probably the authors can clarify on this.
* Given the large volume of the data, it would have been useful for practitioners to provide some insights on the efficiency/scalability of the proposed methods.
* Given the need for encoding long, multilingual documents and retrieve related documents with cosine similarity, some more recent/performant document embeddings might have worked better. For example, GTE embedding (https://huggingface.co/Alibaba-NLP/gte-multilingual-base) is marginally larger than XLM-RoBERTa base (305M vs. 279M), has max token size of 8192 and achieves very good multilingual retrieval performance for its size.

**Strengths Contributions:**

* This is a comprehensive study on different aspects of building a multilingual data curation pipeline.
* The filtering has been proved to significantly improve model performance over the baseline.

---

> ### Author Rebuttal · Authors · 2025-07-30
>
> Dear reviewer Eh93,
>
> We thank you for your positive review and acknowledgement of our work.
>
> **W1:** While various curated multilingual datasets exist, this work aims to add an additional level of quality by improving upon the current state-of-the-art multilingual dataset FineWeb-2. As shown in our work, **our dataset achieves the same performance while training with as little as 15% of the data, making training 6 times faster.** Additionally, high-quality data subselection is often important in later stages of LLM pre-training, as demonstrated by the use of our FineWeb2-HQ dataset in the recently released SmolLM3 model [1].
>
> **W2:** The computation of embeddings required approximately 4K GPU hours. We release the embeddings publicly so this computation does not have to be repeated. Data selection classifier training (i.e. FastText and MLP) can be completed within a few minutes on a CPU. Similarly, cosine similarity can be represented as matrix multiplication and parallelized on a CPU. Filtering is also performed on a CPU and can be easily parallelized. Overall, we spent around 60 CPU hours filtering German, Chinese, French, Arabic, and Danish for one filtering approach in the ablations. Thank you for bringing this to our attention. We will include these additional computational details for filtering in the manuscript.
>
> **W3:** Thank you for pointing out gte-multilingual-base embeddings as alternative. Please note that we specifically selected  XLM-RoBERTa based on our compute budget and its transparent training procedure including the release of the compliant dataset it was trained on.
>
>
> **RW1:** Thank you for highlighting this work. We will incorporate the importance of high-quality training datasets in other domains such as MT.
>
>  **RW2:** Thank you for highlighting this. While this is an interesting avenue of work, we believe that our study applies standard decontamination procedures. A more in-depth discussion on advanced decontamination tools is beyond the scope of our work.
>
> **RW3:** Thank you for highlighting this concurrent work, we will include it in our related work.
>
>
> **Q1:** Thank you for pointing this out. 'Selection' refers to which samples we considered as candidates to train the data selection classifier on, while 'collection' refers to the dataset the classifier was trained on. We will clarify this distinction in the manuscript.
>
> **Q2:** We used 80K samples for each language.
>
> **Q3:** Thank you for pointing this out. The goal was to develop high-quality monolingual datasets, rather than optimizing a multilingual dataset that would additionally be concerned with optimal mixture ratios or might allow for varying qualities among different languages. While we demonstrate that **these refined monolingual datasets also improve multilingual model performance (helping to mitigate the curse of multilinguality)**, our primary focus was on monolingual data quality, which is why we predominantly trained and evaluated monolingual models. We will revise this phrase in the manuscript.
>
> [1] SmolLM3, by Huggingface

---

> ### Comment · Reviewer_Eh93 · 2025-08-01
>
> Thanks for the detailed response. I plan to maintain my score.

---

### Comment · Area_Chair_6mTE · 2025-08-05
**Mandatory Acknowledgement asap**

NeurIPS 2025 Datasets and Benchmarks Track Submission814 Reviewer cncU
NeurIPS 2025 Datasets and Benchmarks Track Submission814 Reviewer dJ3W

Pls submit your Mandatory Acknowledgement asap and provide your feedback!

---

### Decision · Program_Chairs · 2025-09-18

**Decision:**

Accept (poster)

**Comment:**

I think this is a solid dataset paper. The authors propose a simple but effective multilingual data selection framework, show clear gains (same MMLU with ~15% tokens), and also release the data. Reviewers overall leaned positive — some questions on novelty and compute cost, but rebuttal addressed them reasonably. Overall, strengths outweigh weakneses, so I’d go for accept (poster).